# CoIn: Counting the Invisible Reasoning Tokens in Commercial Opaque LLM APIs

## Abstract

As post-training techniques evolve, large language models (LLMs) are increasingly augmented with structured multi-step reasoning abilities, often optimized through reinforcement learning. These reasoning-enhanced models outperform standard LLMs on complex tasks and now underpin many commercial LLM APIs. However, to protect proprietary behavior and reduce verbosity, providers typically conceal the reasoning traces while returning only the final answer. This opacity introduces a critical **transparency gap**: users are billed for invisible reasoning tokens, which often account for the majority of the cost, yet have no means to verify their authenticity. This opens the door to *token count manipulation*, where providers may overreport token usage or inject synthetic, low-effort tokens to inflate charges, a threat that can be carried out at *near-zero cost*. To address this issue, we propose CoIn, a verification framework that audits both the *quantity* and *semantic validity* of hidden tokens. CoIn constructs a verifiable hash tree from token embedding fingerprints to check token counts, and uses embedding-based relevance matching to detect fabricated reasoning content. Experiments demonstrate that CoIn, when deployed as a trusted third-party auditor, can effectively detect token count manipulation with a success rate reaching up to 94.7%, showing the strong ability to restore billing transparency in opaque LLM services. The dataset and code are available at `https://anonymous.4open.science/r/LLM-Auditing-CoIn-20F0`.

## 1 Introduction

Large language models (LLMs) have achieved significant advances in recent years. Yet, as pre-training begins to saturate available data resources (Zoph et al., 2020), the research community has increasingly turned to inference-time innovations (Hu et al., 2023; Kumar et al., 2025). Among these, reinforcement learning (RL)-optimized reasoning models have shown promise by generating longer, structured reasoning traces that improve performance, particularly in tasks involving mathematics and code (Guo et al., 2025; Muennighoff et al., 2025). Such models, exemplified by DeepSeek-R1 (Guo et al., 2025) and ChatGPT-O1 (Jaech et al., 2024), demonstrate that scaling at inference time can yield new capabilities without further pretraining.

With this shift, providers like OpenAI increasingly adopt new service models. Reasoning traces, while critical for quality, are often verbose, sometimes speculative (Jin et al., 2024; Zhang et al., 2025), and may reveal internal behaviors vulnerable to distillation (Gou et al., 2021; Sreenivas et al., 2024). To protect proprietary methods and streamline outputs, commercial APIs typically suppress these intermediate steps, exposing only the final answer. However, users are still charged for all generated tokens, including those hidden from view. We refer to such services as **Commercial Opaque LLM APIs (COLA)**—proprietary, pay-per-token APIs that conceal intermediate reasoning text.

A natural consequence of this design is a **verification gap**: users have no means to verify token usage or detect overbilling. Because reasoning tokens often outnumber answer tokens by more than an order of magnitude (Figure 1), this invisibility allows providers to **misreport token counts** or **inject low-cost, fabricated reasoning tokens to artificially inflate token counts**. We refer to this practice as **token count manipulation**. For instance, a single ARC-AGI run by OpenAI's o3 model consumed 111 million tokens, costing $66,772.[1] Given the scale of reasoning-heavy workloads, even

---

[1] `https://arcprize.org/blog/oai-o3-pub-breakthrough`

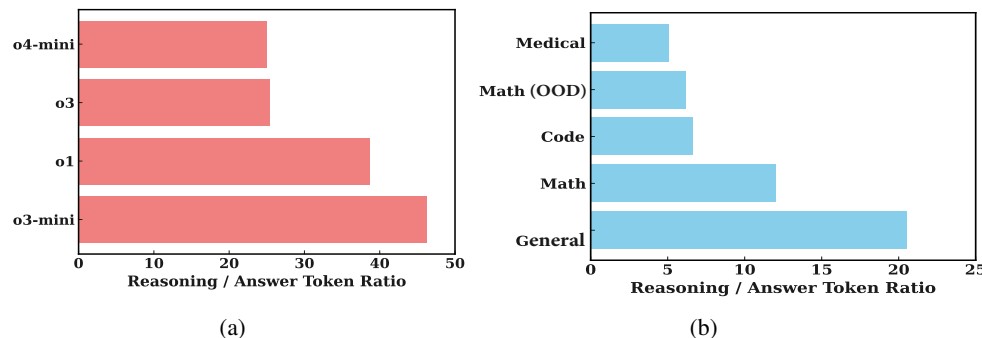

(a)                                           (b)

Figure 1: Ratio of reasoning tokens to answer tokens across datasets and deployed APIs. (a) Token ratios on the OpenR1-Math dataset across different OpenAI reasoning models. (b) Token ratios of the DeepSeek-R1 (Guo et al., 2025) across various reasoning datasets. In both cases, the number of reasoning tokens often exceeds answer tokens by an order of magnitude or more.

small inaccuracies in billing could translate into substantial financial consequences. Although there is no evidence of deliberate misconduct, the asymmetry of information between providers and users underscores the importance of transparent billing mechanisms to safeguard user interests.

To tackle this problem, we design **CoIn (Counting the Invisible)**, a verification framework that enables third-party auditing of invisible reasoning tokens in COLA. ***Importantly, our aim is not to suggest that such practices are occurring in today's systems, but rather to highlight a structural vulnerability inherent to the COLA design.*** On the contrary, we acknowledge the provider's motivations for concealing reasoning traces, as well as the community's concerns about opacity. **CoIn** *seeks to bridge this gap by providing a neutral auditing mechanism that ensures billing accountability while preserving the confidentiality of hidden content.*

CoIn consists of two key components: **(1) Token Quantity Verification**, which leverages a verifiable hash tree (Merkle, 1987) to store fingerprint embeddings of reasoning tokens. Upon an audit request, CoIn allows users to query a small subset of the token fingerprints in the hash tree to verify the number of invisible tokens, avoiding accessing the actual reasoning tokens; and **(2) Semantic Validity Verification**, which detects fabricated, irrelevant, or low-effort token injection via a semantic relevance matching head. This matching head takes the embeddings of both the reasoning tokens and the answer tokens as input, and outputs a relevance score indicating their semantic consistency. Users can assess this score to identify token count manipulation with low-effort token injection. Together, these components enable CoIn to identify misreported token counts and fabricated reasoning traces, enabling transparent billing without exposing proprietary data. In practice, CoIn can be deployed as a trusted third-party auditing service that ensures billing transparency while preserving the integrity and confidentiality requirements of COLA providers.

Our main contributions are as follows:

- We define the COLA architecture and formalize the emerging risk of *token count manipulation*, categorizing it into misreporting, naive inflation, and adaptive inflation strategies.
- We design CoIn, a verification framework combining *token quantity verification* via verifiable hashing and *semantic validity verification* via embedding relevance, to audit invisible tokens without exposing proprietary content.
- Our experiments demonstrate that CoIn can achieve a 94.7% detection success rate against various adaptive attacks with less than 40% embedding exposure and less than 4% token visibility. Moreover, even when 10% of tokens are maliciously forged by COLA, CoIn still maintains a 40.1% probability of successful detection.

## 2 RELATED WORK

**Reasoning Model.** LLMs have shown strong performance on complex reasoning tasks by generating intermediate steps, a technique known as chain-of-thought prompting (Wei et al., 2022). This paradigm has been further enhanced by methods such as self-consistency (Wang et al., 2022) and program-aided reasoning (Gao et al., 2023). Recent research reveals that generating more reasoning steps at inference time can lead to higher answer accuracy, a phenomenon referred to as the test-

time scaling law, which has become a guiding principle for optimizing LLMs (Snell et al., 2024). Reasoning models are typically LLMs fine-tuned via RL (Rafailov et al., 2023; Wu et al., 2023; Ramesh et al., 2024) to produce structured reasoning traces before generating final answers, thereby improving answer quality. These reasoning traces are often longer, more indirect, and may include failed attempts, but are nonetheless closely tied to the final answer (Hao et al., 2024; Yang et al., 2025b). Since these reasoning tokens are generated in the same autoregressive manner as answer tokens, COLAs charge for them based on token count. However, the indirect and verbose nature of reasoning makes it challenging to audit their legitimacy without direct access to the reasoning traces themselves.

**COLA Auditing.** Several works have emerged to address the lack of transparency in COLA. Sun et al. (2025) systematically define the opacity problem of commercial LLM services and further extend it to multi-agent settings. Cai et al. (2025) propose a watermark-based method to audit whether a COLA uses the required LLM rather than a cheaper LLM. Similarly, Yuan et al. (2025) develop a user-verifiable protocol to detect nodes that run unauthorized or incorrect LLM in a multi-agent system. Another series of works (Zheng et al., 2025; Marks et al., 2025) proposes auditing some bad behaviors of LLMs, e.g., cheating and offensive outputs. These techniques mainly focus on the model auditing and lack attention to the token count auditing of COLA.

## 3 PRELIMINARY

**Participants and Problem Formulation.** The `CoIn` framework involves three roles: (1) COLA — a commercial LLM service provider (e.g., OpenAI) that performs multi-step reasoning and returns only the final output to the user; (2) User — an end-user who submits prompts and receives answers along with the billing summary; and (3) `CoIn` auditor — a trusted third party responsible for verifying the invisible reasoning tokens on behalf of the user.

In each service interaction, the user sends a prompt $P$ to COLA. The LLM generates reasoning tokens $R = \{r_1, r_2, \ldots, r_m\}$, followed by answer tokens $A = \{a_1, a_2, \ldots, a_n\}$. Only the final answer $A$ is returned to the user, while the reasoning trace $R$ remains hidden. Billing is based on the total number of tokens $m + n$, including the invisible reasoning tokens. As Figure 1 shows, reasoning tokens often dominate the total count, i.e., $m \gg n$, resulting in a significant transparency gap.

**Potential Token Count Manipulation.** For a malicious COLA, we consider two strategies for token count manipulation:

- **Token Count Misreporting.** COLA reports a falsified token count $m_f > m$, leading to direct overbilling without modifying the output.
- **Token Count Inflation.** Anticipating user-side defenses (e.g., hash matching, spot-checking), COLA may append low-effort fabricated tokens to the original reasoning trace. These fabricated tokens can be generated via random sampling, retrieval from related documents, or repetition of existing tokens, and then indistinguishably mixed with genuine reasoning tokens. The inflated sequence is then used for billing, bypassing naive verification methods and still overcharging the user. Due to the trade-off between risk, benefit, and cost, we only consider ***near-zero cost token inflation***, while fabrications requiring LLM participation are beyond our scope.

To address these threats, `CoIn` employs two components: (1) **Token Quantity Verification**, which audits the reported token count using verifiable commitments and exposes embeddings; and (2) **Semantic Validity Verification**, which evaluates the relevance between reasoning and answer tokens to detect low-quality injections.

**Threat Model.** COLA has access to the user prompt $P$, the full reasoning trace $R$, and the answer $A$, and controls the billing report $(m, n)$, where $m$ is the claimed number of reasoning tokens and $n$ is the number of answer tokens. It can manipulate the reported count without user visibility. The `CoIn` auditor operates as a trusted third party. It can access $P$, $A$, and $(m, n)$, but cannot observe $R$ directly or directly query the LLM used by COLA. However, it can request COLA to return the embeddings of $R$, computed using an embedding model fixed by the auditor to prevent tampering.

## 4 `CoIn`: COUNTING THE INVISIBLE REASONING TOKENS

`CoIn` comprises two complementary components: **token quantity verification** and **semantic validity verification**. The token quantity verification module treats embeddings of invisible reasoning tokens

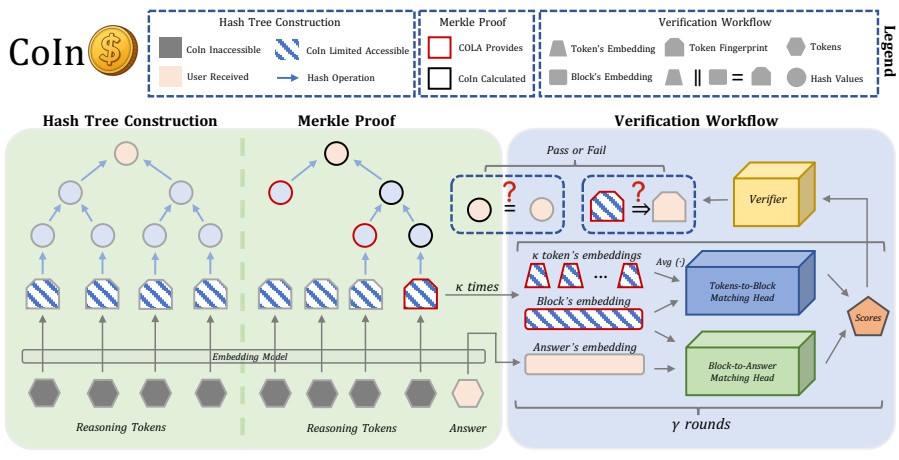

( a ) Token Quantity Verification     ( b ) Semantic Validity Verification

Figure 2: `CoIn` Framework.

as cryptographic fingerprints and organizes them into a verifiable hash tree. By querying a small subset of these fingerprints, users can audit the claimed number of invisible tokens without accessing their contents, thereby mitigating token count misreporting. The semantic validity verification module trains a lightweight neural network, referred to as a *matching head*, to evaluate the relevance between embeddings. During auditing, `CoIn` retrieves token embeddings from the hash tree and uses the matching head to compute relevance scores both among reasoning tokens and between reasoning and answer tokens. These scores help detect token count inflation through the injection of fabricated or irrelevant reasoning tokens. An overview of the `CoIn` framework is illustrated in Figure 2.

## 4.1 TOKEN QUANTITY VERIFICATION

**Token Fingerprint Generation.** In `CoIn`, COLA is required to generate embeddings of its reasoning tokens using a third-party embedding model $\mathsf{Embd}(\cdot)$ designated by the `CoIn` auditor. These embeddings serve as token fingerprints used to construct a verifiable hash tree for auditing. This verifiable hash tree enables `CoIn` to audit the total number of invisible tokens without accessing them, while the per-token hash commitments preclude token count misreporting.

Specifically, given a reasoning sequence $R$, COLA first partitions $R$ into $\alpha$ blocks. For each token $r_i$ in block $B_j$, COLA computes: (1) the block embedding $\mathsf{Embd}(B_j)$, which embeds all the tokens inside the block; and (2) the token embedding $\mathsf{Embd}(r_i)$, which embeds the single token itself. Each reasoning token therefore acquires both the block embedding and the token embedding. For each reasoning token $\mathsf{Embd}(r_i)$, `CoIn` concatenates its block embedding and token embedding to form the token fingerprint: $\mathsf{Embd}(B_j) \,\|\, \mathsf{Embd}(t_i)$.

**Fingerprint Hash Tree Construction.** COLA applies a cryptographic hash function (e.g., SHA-256), agreed upon with `CoIn`, to each token fingerprint to construct the leaf nodes of a Merkle Hash Tree (Merkle, 1987). The number of leaf nodes is padded to the nearest power of two, and parent nodes are built recursively by hashing concatenated sibling nodes up to the Merkle Root. This root serves as a commitment to the full set of reasoning tokens and is submitted to `CoIn`. After constructing the hash tree, COLA gives the Merkle Root to `CoIn` for Merkle Proofs upon user's auditing request.

**Merkle Proof.** Upon receiving the answer $A$ and the token counts $m$ and $n$, a user may suspect token inflation. To verify the count of invisible reasoning tokens, the user selects a block $B_j$ and randomly chooses token indices to audit. Upon receiving the request, `CoIn` auditor requests the following information from COLA: (1) the fingerprints of the selected tokens; and (2) the corresponding Merkle Path, which is a sequence of sibling hashes needed to reconstruct the Merkle Root from the corresponding token. `CoIn` recomputes the Merkle root from the provided data and checks for consistency with the original commitment by COLA. A successful match confirms the integrity of the selected token; a mismatch indicates possible fabrication and inflated token reporting. The construction and Merkle Proof procedure is illustrated in Figure 2-(a) and detailed further in Appendix F.1 , F.2.

The Merkle proof in token quantity verification ensures both the structural integrity and the correctness of the reported token count, effectively defending against token count misreporting. However, a dishonest COLA may still conduct token count inflation by injecting irrelevant or low-effort fabricated tokens that pass count verification. To address this limitation, we introduce semantic validity verification.

## 4.2 SEMANTIC VALIDITY VERIFICATION

To defend against token count inflation, we introduce the semantic validity verification component, as illustrated in Figure 2-(b). This component ensures that reasoning tokens are semantically meaningful and contribute to the final answer, preventing low-effort or fabricated token insertion. Based on this principle, CoIn verifies the semantic validity of invisible tokens from two perspectives:

- **Tokens-to-Block verification** checks whether each reasoning token $r_i$ is semantically coherent within its enclosing block $B_j$. This defends against randomly injected or meaningless tokens.
- **Block-to-Answer verification** evaluates whether a reasoning block $B_j$ is semantically aligned with the final answer $A$, thus identifying the insertion of low-cost content that is insufficiently relevant to the task.

To support both tasks, CoIn trains two lightweight neural modules called the *matching heads*, which are binary classifiers that determine whether two embeddings are semantically associated. Given two token embeddings $a$ and $b$, the matching head first computes the cosine similarity: $\text{cos\_sim} = \frac{a \cdot b}{\|a\|\|b\|}$, and constructs the feature vector: $h = [a; b; a - b; a \odot b; \text{cos\_sim}] \in \mathbb{R}^{4d+1}$, where $d$ is the embedding dimension, $[\,;\,]$ denotes concatenation, and $\odot$ denotes element-wise multiplication. The feature $h$ is then passed through a two-layer feedforward network to produce a scalar match score $S \in [0, 1]$, representing the likelihood that $a$ and $b$ are semantically aligned. This process can be viewed as a regression function $S = \text{MH}(a, b)$.

In CoIn, the matching heads $\text{MH}_{\text{tb}}(\cdot), \text{MH}_{\text{ba}}(\cdot)$ are trained offline for tokens-to-block and block-to-answer verification respectively. CoIn uses open-source corpora and the same embedding model in token fingerprinting to build the datasets for matching heads training.

**Verification Protocol.** In each verification round, the user randomly selects some reasoning tokens $r_i$ (by default, 10% of the tokens within a selected block) from the hash tree. Since the token fingerprint consists of both the token embedding $\text{Embd}(r_i)$ and the corresponding block embedding $\text{Embd}(B_j)$, it can be directly used for Tokens-to-Block verification. For the Block-to-Answer verification, we use $\text{Embd}(B_j)$ and the embedding of the whole answer to compute the score:

$$S_{tb} = \text{MH}_{\text{tb}}(\text{AVG}(\text{Embd}(r_i)), \text{Embd}(B_j)), \qquad S_{ba} = \text{MH}_{\text{ba}}(\text{Embd}(B_j), \text{Embd}(A)). \quad (1)$$

Here, $S_{tb}$ and $S_{ba}$ represent the relevance scores for the two respective verification tasks. Each score reflects the estimated likelihood that the two input embeddings are semantically relevant.

## 4.3 WORKFLOW OF CoIn

**Enforcing Billing Integrity with CoIn.** When a user suspects token count manipulation in a specific response, they can initiate an audit request to CoIn. The audit begins with the user selecting a fraction $\gamma$ of the total reasoning blocks for verification. CoIn then performs two Semantic Validity Verifications and multiple Merkle Proofs on these selected blocks. The resulting match scores are passed to a verifier, which issues a final decision. If the verifier accepts, the audit concludes successfully. If the verifier rejects, the user continues by randomly selecting another unverified block for auditing. This process repeats until either a successful judgment is reached or all blocks are exhausted. If no verification passes, the audit concludes with COLA being flagged for token inflation. The user may then request COLA to justify the charges by disclosing the original reasoning content. The complete procedure is outlined in Algorithm 4.

**Verifier Design.** Each audit round produces a variable-length sequence of match scores, as the number of verified blocks depends on verifier decisions. To handle this, we implement two types of verifiers: (1) **Rule-based**: Averages the scores from two semantic verifications. The audit passes if both averages exceed a threshold $\tau$. (2) **Learning-based**: Uses a lightweight DeepSets model (Zaheer et al., 2017) to process the unordered set of match scores and audit will succeed if the confidence exceeds $\tau$. Auditing outcomes enable users to assess the trustworthiness of a COLA provider.

Frequent failures in `CoIn` audits may erode user trust and damage provider reputation. By introducing verifiable accountability, the `CoIn` framework serves as a deterrent against token count manipulation in commercial LLM services.

**Hyperparameter and Verification Cost.** `CoIn` is governed by a few hyperparameters that control auditing granularity and cost. Specifically, $\alpha$ is the number of blocks, $\beta$ the block size, $\gamma$ the initial sampling ratio (default: 0.3), and $k$ the number of tokens sampled per block (default: $0.1 * \beta$). A smaller $\beta$ reduces exposure but increases overhead. The protocol begins with $\gamma \cdot \alpha$ rounds and may proceed up to $\alpha$ rounds under early stopping, so the number of verification rounds satisfies $\ell \in [\gamma \cdot \alpha, \ \alpha]$. As a result, the total number of Merkle Proofs is $k \cdot \ell$, and the number of Semantic Judgments is $2 \cdot \ell$.

## 5 EXPERIMENTS

We systematically evaluate the robustness and reliability of `CoIn` and its submodules under various adaptive inflation attacks across multiple datasets. We further analyze the construction cost of the Hash Tree, as well as whether the partially exposed block embeddings and tokens can be exploited to recover the reasoning tokens of COLA. Finally, we assess the difficulty of the dataset we constructed.

### 5.1 EXPERIMENT SETUP

**Token Inflation Implementations.** We study both *naive* and *adaptive* token count inflation strategies. To enable fine-grained evaluation and systematic dataset construction, we design four variants of adaptive inflation. All inflation types used in our experiments are summarized in Table 1. These strategies are applied to generate inflated samples for both training and evaluation.

Table 1: Token inflation types used in our experiments.

| Type | Description |
| --- | --- |
| **Naive Inflation** | Randomly select tokens from the vocabulary for injection. |
| **Ada. Inflation 1** | Inject tokens with embeddings similar to $P$, $R$, or $A$. |
| **Ada. Inflation 2** | Inject tokens directly sampled from $P$, $R$, or $A$. |
| **Ada. Inflation 3** | Inject reasoning sequences extracted from other inputs. |
| **Ada. Inflation 4** | Inject retrieved sequences semantically similar to $P$, $R$, or $A$. |

**Datasets and Training Setup.** We conduct experiments on five datasets derived from `DeepSeek-R1` (Guo et al., 2025), covering diverse reasoning domains: medical (Chen et al., 2024a), code (Team, 2025; Face, 2025), mathematics (Face, 2025), general reasoning[2], and out-of-domain (OOD) mathematics (Team, 2025; Face, 2025). For training, we randomly sample 20,000 examples from each dataset and combine them into a joint dataset. Another 1,000 samples per dataset are held out to form the evaluation set for `CoIn`. We use the tokenizer of `DeepSeek-R1` in our experiments.

For the matching head, we use `all-MiniLM-L6-v2` (Reimers & Gurevych, 2019) as model structure. In the **tokens-to-block** verification task, we treat original samples as normal instances and apply Naive Inflation as well as Adaptive Inflation 1 and 2 to construct inflated samples. Normal and inflated samples are labeled 0 and 1 respectively and mixed at a 1:1 ratio to form the training set. For the **block-to-answer** verification task, we adopt a similar setup, but use Ada. Inflation 1–4 to construct inflated samples. This setting enables a thorough evaluation of the model's ability to detect both shallow and semantically sophisticated inflation attacks. The details are explained in Appendix C, D.

**Metrics.** We define the *Detection Success Rate (DSR)* as the classification accuracy of our module, computed separately for malicious and benign samples. Unlike conventional metrics (e.g. AUC), DSR highlights performance differences across negative sample types and reveals asymmetric detection difficulty that aggregated metrics may obscure. *Inflation Rate (IR)* indicates the percentage of fabricated tokens injected by COLA relative to the number of original reasoning tokens. For benign samples, the *Average Exposure Rate (AER)* refers to the proportion of blocks exposed during the `CoIn` multi-step verification process out of the total number of blocks.

---

[2]https://huggingface.co/datasets/glaiveai/reasoning-v1-20m

## 5.2 DETECTION PERFORMANCE OF `CoIn`

We evaluate `CoIn`'s ability to detect various token count inflation attacks. Figure 3 shows the relationship between IR and DSR across five datasets. DSR increases with IR, indicating that more aggressive inflation is easier to detect. For example, in the Naive Inflation setting, DSR approaches 100% when IR exceeds 0.5, suggesting near-perfect detection for heavily inflated queries. Figure 4 illustrates the impact of $\tau_h$. As $\tau_h$ increases, DSR for malicious samples increases, while for benign samples decreases. This highlights a tunable trade-off between user utility and COLA's protection.

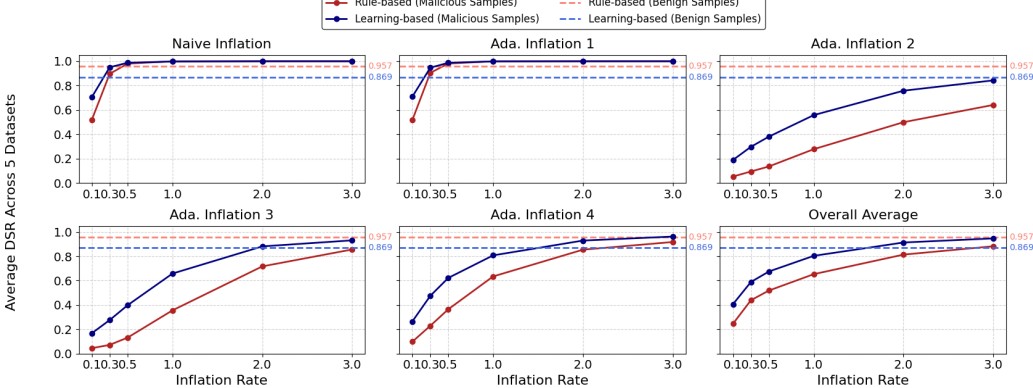

Figure 3: Performance of `CoIn` across different inflation methods and verifiers. The red lines and the blue lines represent the DSR of rule-based verifier and learning-based verifier, respectively.

**Learning-based Verifier Excels at Detecting Malicious Samples.** In Figure 3, for a fair comparison, we set the threshold $\tau = 0.5$ for learning-based verifier, and $\tau = 0.6$ for rule-based verifier. The initial sampling ratio $\gamma$ is set to 0.3. Across all settings, the learning-based verifier consistently outperforms the rule-based variant in detecting inflated samples, achieving up to 94.7% average DSR at an inflation ratio (IR) of 3.0. Among the adaptive methods, Ada. Inflation 2 and Ada. Inflation 3 present greater detection challenges. Nevertheless, `CoIn` still achieves an average DSR of approximately 84.3% and 93.1% at IR = 3.0 for these cases. These results demonstrate that `CoIn` remains robust even under strong adaptive inflation, and that learning-based semantic verification offers significant advantages in practical detection scenarios.

Table 2: Influence of Block Size.

| Metric | Block Size $\beta$ | | |
|---|---|---|---|
| | 256 | 512 | 1024 |
| Avg. $\alpha$ | 16.8 | 8.6 | 4.5 |
| Avg. $\ell$ | 6.3 | 3.7 | 2.2 |
| AER $\downarrow$ | 0.38 | 0.43 | 0.49 |

**Rule-based Verifier Excels at Handling Benign Samples.** Although it is less effective at detecting inflated samples, the rule-based verifier, as shown in Figure 3, performs better on benign samples when properly tuned, due to its lower false positive rate. Given our verification mechanism, a higher DSR on benign samples results in a lower Average Exposure Rate (AER), which makes COLA more favorable toward such mechanisms. Based on the rule-based verifier ($\tau = 0.6$), Table 2 reports the average number of blocks $\alpha$, average verification rounds $\ell$ and AER across different block sizes $\beta$. With increasing $\beta$, $\ell$ decreases, indicating lower cost. However, AER increases, and the results in Appendix G show that increasing the block size reduces the DSR for adaptive inflation.

## 5.3 PERFORMANCE OF THE SEMANTIC VALIDITY VERIFICATION

**Block-to-Answer Verification.** We separately evaluate the performance of the two types of matching heads introduced in Section 4.2. Table 3 shows the DSR of the matching heads on the *Block-to-Answer Verification* task. The model achieves an average DSR of 94.8% across attack types. Even for the Math (OOD) dataset, which was excluded from training, the model performs strongly, indicating good generalization. The DSR drops slightly on clean (non-inflated) samples due to the presence of reasoning blocks not directly contributing to the final answer (see Section 6). Additionally, Ada. Inflation 3 introduced hard negatives that resemble real data, making separation more difficult.

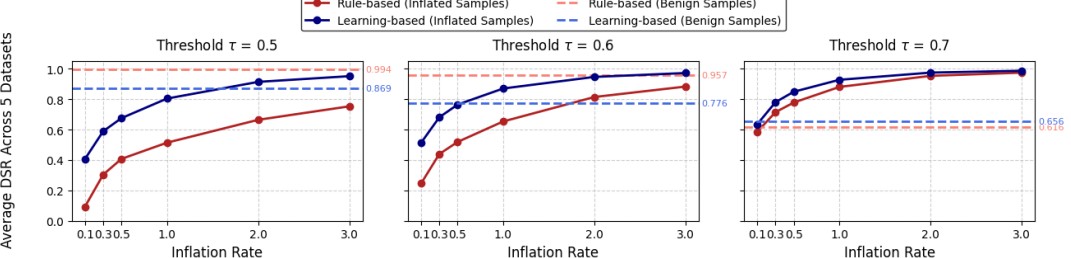

Figure 4: Impact of threshold $\tau$ on DSR.

Table 3: Block-to-Answer Verification Performance Across Attack Types and Domains.

| Attack Type | Medical | Code | Math | General | Math (OOD) | Avg. |
|---|---|---|---|---|---|---|
| **Naive Inflation** | 99.4 | 100.0 | 100.0 | 99.3 | 100.0 | 99.7 |
| **Ada. Inflation 1** | 95.3 | 98.7 | 98.6 | 96.8 | 98.2 | 97.5 |
| **Ada. Inflation 2** | 94.4 | 92.3 | 92.8 | 94.2 | 92.7 | 93.3 |
| **Ada. Inflation 3** | 89.2 | 81.5 | 84.3 | 92.9 | 84.6 | 86.5 |
| **Ada. Inflation 4** | 94.2 | 97.9 | 99.0 | 96.1 | 97.8 | 97.0 |
| **Avg. With Inflation** | 94.5 | 94.1 | 94.9 | 95.8 | 94.7 | 94.8 |
| **No Inflation** | 87.9 | 90.3 | 87.1 | 86.5 | 87.9 | 87.9 |

**Tokens-to-Block Verification.** Table 4 shows the results for tokens-to-block verification. The model performs well overall but struggles with Adaptive Inflation 2, where tokens reused from the same sample lead to significant lexical and semantic overlap. This overlap can blur the distinction between original and fabricated content, especially when reused tokens legitimately contribute to the block.

Table 4: Tokens-to-Block Verification Performance Across Attack Types and Domains.

| Attack Type | Medical | Code | Math | General | Math (OOD) | Avg. |
|---|---|---|---|---|---|---|
| **Naive Inflation** | 90.8 | 90.5 | 95.3 | 84.5 | 94.6 | 91.2 |
| **Ada. Inflation 1** | 95.1 | 96.1 | 95.8 | 95.5 | 95.8 | 95.6 |
| **Ada. Inflation 2** | 76.0 | 75.2 | 73.9 | 73.6 | 74.8 | 74.7 |
| **Avg. With Inflation** | 87.3 | 87.2 | 88.4 | 84.5 | 88.4 | 87.2 |
| **No Inflation** | 82.0 | 80.4 | 87.2 | 79.0 | 86.0 | 82.9 |

**Cost of Building Hash Trees.** We evaluate the computational overhead of constructing the Merkle Tree, with respect to input size and hidden dimension. Experiments were conducted on a dual-socket AMD EPYC 7763 system (128 cores, 256 threads). All constructions ran as single-threaded processes on one logical core. As shown in Figure 5, the construction time grows approximately linearly with the input length for a fixed hidden dimension, and increases more steeply with higher dimensions. Given that most LLM inference servers have underutilized CPUs, and the Merkle Tree construction process scales effectively with parallelism, the cost of building it is nearly negligible.

## 6 DISCUSSION

**Can the original text be recovered from the tokens and embeddings exposed by COLA?** During the `CoIn` verification process, COLA may leak partial block embeddings and tokens to `CoIn`. To quantify the impact of such leakage, we design two experiments: (1) **Direct reconstruction from embeddings**. Our protection goal is to prevent large-scale reasoning data from being collected for reverse engineering or distillation, rather than avoiding leakage of individual reasoning fragments. We adopt Vec2Text (Morris et al., 2023), a "hypothesis–correction" iterative method for embedding inversion. As shown in Table 5, text can still be partially recovered when the block size is small ($\leq 64$), but recovery performance drops sharply as block size increases; at our framework's minimum block size of 256, the attack almost completely fails. Even when BLEU (Papineni et al., 2002) / F1 scores are relatively high, the reconstructed text often suffers from severe semantic distortion, making

it unsuitable for malicious data distillation. (2) **LLM-based reconstruction**. We further assume that a malicious `CoIn` may employ RAG to retrieve similar documents and use an LLM to reconstruct the original content (prompt design provided in Appendix H). Results on a math dataset (Table 7) show that the combination of high BERTScore/EmbedSim and low BLEU/ROUGE indicates that LLMs can preserve core semantics, but differ significantly in surface expression and syntactic structure. Further experimental details are provided in Appendix I.

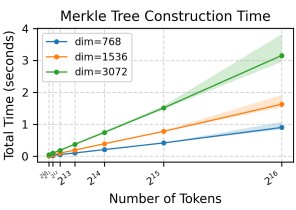

Figure 5: Merkle Tree Construction Time with Fluctuation Range.

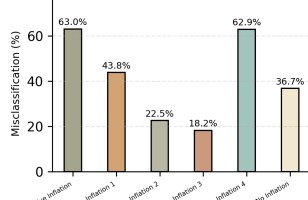

Figure 6: Misclassification Rates of LLMs on Constructed Datasets.

Table 5: Reconstruction performance on Math (see full results in Appendix I.1, Table 6).

| Block Size | Math | |
|---|---|---|
| | BLEU | Token F1 |
| 16 | 43.83 | 0.7467 |
| 32 | 15.06 | 0.5123 |
| 64 | 12.62 | 0.4256 |
| 128 | 9.84 | 0.3441 |
| 256 | 4.07 | 0.2617 |
| 512 | 0.67 | 0.2016 |

**How does `CoIn` defend against repetition-based token inflation?** An important class of adversarial strategies we must consider involves dishonest providers artificially inflating token counts by repeating reasoning segments or appending permuted or LLM-rewritten variants. For simplicity, we did not incorporate such strategies into the core method presented in this paper, but `CoIn` can be naturally extended to mitigate them. For direct repetition, subtree equivalence checks in the Merkle tree can reveal duplicated segments. *If repeated content exists, identical hash values will appear at sibling nodes, implying structurally identical subtrees and thus duplicated token sequences.* For more complex manipulations, such as random permutations or LLM-based rewritings, we provide detailed experiments in Table 8. We find that even after rewriting by Qwen3-4B (Yang et al., 2025a), the rewritten text still exhibits high similarity to the original. Therefore, the proposed subtree similarity check retains strong potential to detect such inflation patterns. As a highly extensible framework, `CoIn` can incorporate these additional measures to defend against a broader range of attacks. Detailed analysis and experimental results are provided in Appendix J.

**How difficult is the dataset we constructed?** To investigate the dataset difficulty, we submitted the failed samples from the *Block-to-Answer Verification* task, along with their Answer, to a LLM. Based on the idea of LLM-as-a-Judge (Zheng et al., 2023; Li et al., 2024), we use a prompt to instruct the LLM to perform binary classification. The prompt used is provided in Appendix H. The relatively high misclassification rate suggests that the LLM, after reading the original text, tends to align with the matching head's judgment. The LLM shows high error rates on Naive Inflation, Ada. Inflation 1 and 4, indicating strong performance of the matching head in these cases. However, it still struggles with the remaining two adaptive inflations. Notably, 36.7% of real blocks were misclassified by the LLM, suggesting that some parts of the true reasoning steps may be unrelated to answer derivation.

## 7 CONCLUSION

This paper presents `CoIn`, a novel auditing framework designed to verify the token counts and semantic validity of hidden reasoning traces in COLA. We identify and formalize the problem of *token count manipulation*, in which service providers can overcharge users by injecting redundant or fabricated reasoning tokens that are not visible to the user, often at *near-zero computational cost*. To address this, `CoIn` integrates two complementary components: a hash tree-based token quantity verifier and a semantic relevance-based validity checker. Our extensive experiments demonstrate that `CoIn` can detect both naive and adaptive inflation strategies with high accuracy, even under limited exposure settings. By enabling transparent and auditable billing without revealing proprietary content, `CoIn` introduces a practical mechanism for accountability in commercial LLM services. We hope this work lays the foundation for future research on LLM API auditing, transparent reasoning, and verifiable inference services.

ETHICS STATEMENT

The central goal of this research is to enhance billing transparency and accountability in commercial opaque LLM APIs, thereby fostering greater trust between service providers and users. Our work identifies a potential vulnerability, *token count manipulation*, and proposes a defensive framework, CoIn, to address it. We stress that our position is entirely neutral: we are not suggesting, implying, or accusing any current commercial providers of engaging in such practices. Instead, we view this research as a proactive exploration of potential risks that could arise from information asymmetries between providers and users. Our objective is to contribute constructively to the ecosystem by identifying possible vulnerabilities early and proposing mitigations that help prevent the erosion of trust.

Ultimately, we believe this work contributes positively to the AI ecosystem by introducing a mechanism that balances the provider's need to protect intellectual property with the user's right to verifiable billing. We therefore believe this research raises no significant ethical concerns beyond the general considerations outlined in the ICLR Code of Ethics.

REPRODUCIBILITY STATEMENT

To ensure the reproducibility of our research, we have made our code and datasets publicly available under an anonymous license. The complete implementation of the CoIn framework, along with the scripts used for all experiments, can be found at the following anonymous repository:

https://anonymous.4open.science/r/LLM-Auditing-CoIn-20F0.

We provide extensive details of our experimental setup in the paper, specifically:

- **Dataset Construction:** The methodology for creating the evaluation and training datasets, including various token inflation strategies, is detailed in Section 5.1. Additional details, such as data sources with their corresponding HuggingFace links, dataset construction methods, and data-related experimental settings, are further discussed in Appendix C. Due to file size limitations, we provide partial data samples through an anonymous link; the complete dataset and the trained models involved will be released after the paper is accepted.
- **Model and Training:** The overall framework design, the architecture of the matching head, and its execution method are discussed in Section 4. Hyperparameters for training the matching heads and the verifier, as well as detailed model architecture descriptions, are provided in Appendix D.
- **Algorithms:** The core algorithms for Merkle Tree construction, Merkle Proof verification, and the complete multi-round CoIn workflow are formally described in Algorithms 2, 3, 4 in Appendix F.

Although we used LLMs to draft some data preprocessing scripts, all code has been manually reviewed and verified by the authors. Furthermore, algorithmic skeletons generated by LLMs were based on original Python code provided by the authors, and subsequently refined with additional details by hand. The final released codebase ensures complete reproducibility. We believe these resources provide a solid foundation for other researchers to verify our results and build upon our work.

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

## A  LLM USAGE STATEMENT

In the development of this work, we employed several commercial LLMs at different stages to enhance the quality and robustness of our research. The specific applications are as follows:

**Adversarial Brainstorming.** After establishing the main framework of `CoIn`, we interacted with LLMs in an adversarial manner by prompting them to act as attackers and propose potential exploits and attack strategies against our design. This iterative process played an important role in identifying weaknesses in the framework and improving our defense mechanisms. For example, adversarial brainstorming inspired the idea of a multi-round verification mechanism: in cases of verification failure, the `CoIn` framework could require a COLA to provide the original text, which substantially increases the risk of fraudulent behavior being exposed. At the same time, we carefully considered and ultimately rejected certain potential directions, such as zero-knowledge-proof-based defenses, primarily due to concerns about practicality and scope.

**Partial Code Generation and Code-to-Algorithm Conversion.** We employed LLMs to assist with technical implementation. For instance, LLM was used to draft Python scripts for data preprocessing. For some of the more complex algorithms in the appendix, we provided original Python code as guidance and prompted LLMs to generate high-level LaTeX algorithmic skeletons based on this code. These outputs then served as blueprints, which were thoroughly reviewed, refined, and supplemented by the authors to ensure correctness and methodological consistency.

**Manuscript Polishing.** We used LLMs as writing assistants to improve sentence structure, check spelling errors, and enhance the clarity and readability of the manuscript.

All LLM-generated content (including conceptual challenges, code, and text) was critically reviewed by the authors, who take full responsibility for the scientific integrity and accuracy of the paper. We emphasize that LLMs are not authors and bear no responsibility for this work; all accountability lies with the human authors. This study did not involve any human subjects or sensitive data, and therefore raises no additional ethical concerns beyond those discussed in our Ethics Statement.

## B  LIMITATIONS

We acknowledge that `CoIn`, despite its merits, possesses certain limitations that warrant discussion.

- **Mechanistic limitations**: When the inflation rate is low, `CoIn` shows limited performance in detecting malicious samples; its probabilistic nature also inevitably leads to a non-zero misclassification rate. In cases where benign samples are misclassified as malicious, the protocol requires COLA to disclose the original text for verification. Furthermore, the auditing process of `CoIn` depends on COLA's active cooperation, which may constrain its applicability in practice.
- **Uncovered attack surfaces**: `CoIn` remains less effective against certain sophisticated attacks. For instance, LLM-based rewriting or expansion attacks may bypass detection, but they require substantial computational cost, far beyond the near-zero cost scenarios considered in this work, and are thus better categorized as model substitution Cai et al. (2025); Chen et al. (2024b), for which a separate line of research exists and could be combined with `CoIn`. Moreover, cascade-based inference Liao et al. (2025) or speculative sampling Chen et al. (2023) poses ambiguous cases of "inflation," where it is difficult to distinguish fraud from legitimate reasoning optimization, leaving this aspect unverified.
- **Potential privacy risks**: Although Section 6 shows that recovering original tokens from embeddings Morris et al. (2023) under `CoIn`'s setting is difficult, future advances or attacks tailored specifically to our setting may still expose partial information. While such partial leakage is unlikely to enable the construction of high-quality distillation datasets, it may become a serious concern in sensitive domains such as healthcare.

## C DATASET CONSTRUCTION AND EXPERIMENTAL DETAILS

### C.1 DATASET CONSTRUCTION DETAILS

We construct two verification datasets for Block-to-Answer and Tokens-to-Block verification, each dataset includes two types of inflated samples. The simple version consists entirely of artificially generated (inflated) tokens, while the hard version contains a mixture of real and inflated tokens. For Tokens-to-Block verification, we randomly sample between 3.125% and 12.5% of tokens from each block to create both training and test instances.

For both verification tasks, we generate 1,200,000 positive and negative samples respectively. The training set is uniformly distributed across four datasets. Since the difficulty levels of the samples vary, we adjust the composition using an adaptive inflation strategy (applied in Block-to-Answer) to ensure balanced learning.

For training the DeepSets model, we additionally sample 1,000 examples. To preserve generalization capability, the data used for training this model does not overlap with any samples seen by the matching heads.

### C.2 EXPERIMENTAL DETAILS

All evaluation results, unless stated otherwise, are reported on 1,000 examples. This applies to Block-to-Answer, Tokens-to-Block, and the test sets used within the `CoIn` framework. Each numeric result is computed over a minimum of 1,000 samples to ensure statistical significance. Please refer to the Algorithm 1 for our `CoIn` workflow test set construction process.

---

**Algorithm 1** Streamlined Generation of Inflated Reasoning Sequences

---

**Require:** Original dataset $D_{orig}$, inflation ratios $\mathcal{K}$, strategies $S_{list}$ with weights $W_S$, tokenizer $\mathcal{T}$, embedder $\mathcal{E}$, anchor source $Src_{anchor}$, segment length range $[L_{min}, L_{max}]$, insertion mode $M_{ins}$, and optional block range $[B_{min}, B_{max}]$ if using block mode.
**Ensure:** Inflated dataset $D_{inflated}$
1: Initialize $D_{inflated} \leftarrow \emptyset$
2: Build FAISS indexes for RAG-based strategies
3: **for** each data point $item_i = (P_i, R_i, A_i)$ in $D_{orig}$ **do**
4:     $T_{orig} \leftarrow \mathcal{T}(R_i)$;
5:     **if** $T_{orig}$ is empty **then continue**
6:     **end if**
7:     $T_{anchor} \leftarrow \text{SelectAnchor}(item_i, Src_{anchor})$
8:     $N_{max} \leftarrow \lfloor |T_{orig}| \cdot \max(\mathcal{K}) \rfloor$
9:     $T_{pool} \leftarrow \text{CollectTokens}(N_{max}, T_{anchor}, S_{list}, W_S)$
10:     **for** each $k \in \mathcal{K}$ **do**
11:         $N_k \leftarrow \lfloor |T_{orig}| \cdot k \rfloor$
12:         $T_k \leftarrow \text{Subsample}(T_{pool}, N_k)$
13:         $T_{final} \leftarrow \text{Insert}(T_{orig}, T_k, M_{ins}, [B_{min}, B_{max}])$
14:         Add $\mathcal{T}^{-1}(T_{final})$ to $D_{inflated}$ with metadata
15:     **end for**
16: **end for**
17: **return** $D_{inflated}$

---

### C.3 TOKEN INJECTION STRATEGY FOR EVALUATION DATASET

To further clarify our dataset construction, we describe the token injection process used in generating evaluation samples. For each original reasoning sequence, we apply the following procedure:

- **Compute required malicious tokens.** The number of injected tokens is determined by a predefined inflation rate, ranging from 10% to 300% of the original reasoning length.
- **Sample malicious tokens.** Depending on the chosen inflation method, we collect sufficient malicious tokens. For continuous text segments (e.g., Ada. Inflation 4, sampled from Wikipedia), we ensure that each sampled document is between 256–512 tokens in length.

- **Partition into malicious blocks.** The malicious tokens are grouped into blocks of length between 32–256 tokens.
- **Random insertion.** Each malicious block is randomly inserted into the original reasoning content, leading to stochastic placement across different reasoning positions.
- **Final block partition.** After injection, the reasoning content (now containing malicious blocks) is split into fixed-size blocks according to the block size $\beta$.

This injection strategy creates more challenging and realistic evaluation cases. Because the placement of malicious blocks is random, some reasoning blocks may contain no malicious content, while others contain only 10–20% malicious tokens. Such variability increases the difficulty of detection and ensures robustness of the evaluation.

### C.4 SOURCE OF DATASET

To evaluate `CoIn`'s performance across different domains, we constructed training and test sets based on five datasets distilled from `DeepSeek-R1` Guo et al. (2025), including Medical Chen et al. (2024a)[3], Code Team (2025); Face (2025)[4], Math Face (2025)[5], General[6], and Out-of-Domain data Math (OOD) Team (2025); Face (2025)[7]. Our final training set is a mixture of these five datasets.

## D TRAINING AND MODEL DETAILS

### D.1 HYPERPARAMETERS

For the matching heads used in Tokens-to-Block verification and Block-to-Answer verification, we set the learning rate to $2 \times 10^{-5}$, the batch size to 128, and train for 3 epochs. We employ the Adam optimizer and use the focal loss function. The hidden dimension of the model follows that of the embedding model, set to 384.

For the DeepSets model in the verifier, we use a batch size of 128, a hidden dimension of 256, and train for 5 epochs. We adopt the Adam optimizer with a learning rate of $1 \times 10^{-3}$ and use the binary cross-entropy (BCE) loss. All experiments are conducted with a fixed random seed of 42.

### D.2 TRAINING OF MATCHING HEADS

As described in the main text, `MH_tb` and `MH_ba` are two lightweight neural matching heads that take as input a pair of embeddings and output a score between 0 and 1, indicating their semantic relevance. Internally, these modules extract interaction features and feed them into two-layer feed-forward networks. Below we detail how we construct diverse training pairs for both modules.

**Block-to-Answer Matching (MH_ba).**

- **Benign samples:** Extract continuous segments from the original reasoning blocks.
- **Malicious samples:** Cover all inflation types discussed in the paper:
  1. *Naive Inflation:* Tokens randomly sampled from the vocabulary.
  2. *Adaptive Inflation 1:* Embed the prompt, reasoning, and answer, then inject semantically similar tokens based on cosine similarity.
  3. *Adaptive Inflation 2:* Tokens or phrases randomly sampled from the prompt/reasoning/answer and re-injected into the reasoning.
  4. *Adaptive Inflation 3:* For a given sample, find the most similar sample (via embedding) in the dataset and inject its reasoning into the current one.
  5. *Adaptive Inflation 4:* Similar to Inflation 1, but retrieves full paragraphs from an external corpus (e.g., Wikipedia) based on embedding similarity.

---

[3]https://huggingface.co/datasets/FreedomIntelligence/Medical-R1-Distill-Data
[4]https://huggingface.co/datasets/open-r1/OpenThoughts-114k-Code_decontaminated
[5]https://huggingface.co/datasets/open-r1/OpenR1-Math-220k
[6]https://huggingface.co/datasets/glaiveai/reasoning-v1-20m
[7]https://huggingface.co/datasets/open-r1/OpenThoughts-114k-math

To ensure difficulty, the ratio of real reasoning to injected tokens is uniformly controlled in the $[20\%, 80\%]$ range. We also construct fully malicious samples where all blocks are from inflation sources, maintaining a 1:1 ratio between partial and full malicious samples.

**Tokens-to-Block Matching (MH_tb).**   This module is trained using the same dataset as block-to-answer. Instead of matching entire blocks, we randomly select 5–20 tokens from the original text, average their embeddings, and use this token-group embedding as input alongside the block embedding.

**Dataset Size and Balance.**   For both tasks, we balance benign and malicious samples, with 1.2 million samples per class.

Due to space limitations, these details could not be included in the main text. They are now provided in Appendix D.2, alongside code in the supplementary material.

## E   COMPUTATIONAL RESOURCES

All experiments were conducted on a high-performance workstation running Ubuntu 20.04.6 LTS. The system is equipped with a dual-socket AMD EPYC 7763 processor, providing a total of 128 physical cores and 256 threads. For GPU acceleration, we utilized an NVIDIA RTX A6000 Ada graphics card.

## F DETAILS OF CoIn

### F.1 MERKLE TREE CONSTRUCTION

Algorithm 3 details the process COLA uses to construct the Merkle Hash Tree from a reasoning sequence $R$. This corresponds to the "Token Fingerprint Generation" and "Fingerprint Hash Tree Construction" paragraphs.

### F.2 MERKLE PROOF VERIFICATION

Algorithm 2 describes how the CoIn auditor verifies the integrity of a token using its fingerprint and the Merkle path provided by COLA. This corresponds to the "Merkle Proof" paragraph.

---

**Algorithm 2** Merkle Proof Verification

---

**Require:** Committed Merkle Root $MR_{committed}$ (from COLA).
**Require:** Token fingerprint $fp_{token}$ of the audited token (from COLA).
**Require:** Merkle Path $P = [(h_1, pos_1), (h_2, pos_2), \ldots, (h_d, pos_d)]$ (from COLA), where $h_k$ is a sibling hash and $pos_k \in \{\text{'left', 'right'}\}$ indicates $h_k$'s position relative to the path node.
**Require:** Cryptographic hash function $H(\cdot)$.
**Ensure:** Boolean: **true** if verification succeeds, **false** otherwise.
 1: $current\_computed\_hash \leftarrow H(fp_{token})$         ▷ Hash the provided token fingerprint
 2: **for** each pair $(sibling\_hash, position) \in P$ **do**
 3:     **if** $position = \text{'left'}$ **then**
 4:         $current\_computed\_hash \leftarrow H(sibling\_hash \,\|\, current\_computed\_hash)$
 5:     **else if** $position = \text{'right'}$ **then**
 6:         $current\_computed\_hash \leftarrow H(current\_computed\_hash \,\|\, sibling\_hash)$
 7:     **else**
 8:         **return false**         ▷ Error: Invalid position in Merkle Path
 9:     **end if**
10: **end for**
11: $MR_{recomputed} \leftarrow current\_computed\_hash$
12: **if** $MR_{recomputed} = MR_{committed}$ **then**
13:     **return true**         ▷ Verification successful: token integrity confirmed
14: **else**
15:     **return false**         ▷ Verification failed: mismatch indicates potential issue
16: **end if**

---

**Notes on Algorithms:**

- **Padding (Algorithm 3):** The text states, "The number of leaf nodes is padded to the nearest power of two." Algorithm 3 implements this by duplicating the hash of the last actual leaf node if leaves exist. If the initial set of tokens (and thus fingerprints) is empty ($N = 0$), it assumes padding to $N_{pow2} = 1$ using a hash of a predefined value (e.g., an empty string). The exact nature of this padding for an empty set should be consistently defined between COLA and the auditor.
- **Merkle Path Representation (Algorithm 2):** The Merkle Path $P$ is assumed to be a list of (hash, position) tuples. The 'position' indicates if the sibling hash is to the 'left' or 'right' of the node on the direct path from the audited leaf to the root.
- **Concatenation for Hashing:** The order of concatenation (e.g., $H(leftChild \,\|\, rightChild)$ vs. $H(rightChild \,\|\, leftChild)$) must be consistent throughout construction and verification. The algorithms assume a fixed order (left child first).

---

**Algorithm 3** Merkle Tree Construction by COLA

---

**Require:** Reasoning tokens $R$; number of blocks $\alpha$; embedding function $\mathsf{Embd}(\cdot)$; cryptographic hash function $H(\cdot)$.

**Ensure:** Merkle Root $MR$.

    *// Phase 1: Token Fingerprint Generation and Leaf Node Creation*

1: $Blocks \leftarrow \mathsf{Partition}(R, \alpha)$                                         $\triangleright$ Partition $R$ into $B_1, \ldots, B_\alpha$

2: $Fingerprints \leftarrow \emptyset$                                    $\triangleright$ Initialize as an empty list

3: **for** each block $B_j \in Blocks$ **do**

4:     $e_{block_j} \leftarrow \mathsf{Embd}(B_j)$                           $\triangleright$ Compute block embedding

5:     **for** each token $r_i \in B_j$ **do**

6:         $e_{token_i} \leftarrow \mathsf{Embd}(r_i)$                     $\triangleright$ Compute token embedding

7:         $fp_i \leftarrow e_{block_j} \,\|\, e_{token_i}$                 $\triangleright$ Form token fingerprint

8:         Add $fp_i$ to $Fingerprints$

9:     **end for**

10: **end for**

11: $LeafNodes \leftarrow \emptyset$                                 $\triangleright$ Initialize as an empty list

12: **for** each fingerprint $fp \in Fingerprints$ **do**

13:     $leaf \leftarrow H(fp)$                     $\triangleright$ Hash fingerprint to create leaf node

14:     Add $leaf$ to $LeafNodes$

15: **end for**

    *// Phase 2: Padding Leaf Nodes*

16: $N \leftarrow \mathrm{length}(LeafNodes)$

17: Let $N_{pow2}$ be the smallest power of two such that $N_{pow2} \geq N$.

18: **if** $N < N_{pow2}$ **then**

19:     **if** $N = 0$ and $N_{pow2} > 0$ **then**              $\triangleright$ e.g., $N = 0 \implies N_{pow2} = 1$

20:         $padding\_hash \leftarrow H(\texttt{""})$    $\triangleright$ Hash of empty string or other predefined padding value

21:         **for** $k \leftarrow 1$ **to** $N_{pow2}$ **do**

22:             Add $padding\_hash$ to $LeafNodes$

23:         **end for**

24:     **else if** $N > 0$ **then**

25:         $last\_leaf\_hash \leftarrow LeafNodes[N-1]$         $\triangleright$ Get hash of the last actual leaf

26:         **for** $k \leftarrow 1$ **to** $N_{pow2} - N$ **do**

27:             Add $last\_leaf\_hash$ to $LeafNodes$     $\triangleright$ Pad by duplicating the last leaf's hash

28:         **end for**

29:     **end if**

30: **end if**

    *// Phase 3: Building the Tree Recursively*

31: $CurrentLevelNodes \leftarrow LeafNodes$

32: **while** $\mathrm{length}(CurrentLevelNodes) > 1$ **do**

33:     $NextLevelNodes \leftarrow \emptyset$

34:     **for** $k \leftarrow 0$ **to** $(\mathrm{length}(CurrentLevelNodes)/2) - 1$ **do**

35:         $leftChild \leftarrow CurrentLevelNodes[2k]$

36:         $rightChild \leftarrow CurrentLevelNodes[2k+1]$

37:         $parentHash \leftarrow H(leftChild \,\|\, rightChild)$

38:         Add $parentHash$ to $NextLevelNodes$

39:     **end for**

40:     $CurrentLevelNodes \leftarrow NextLevelNodes$

41: **end while**

42: **if** $\mathrm{length}(CurrentLevelNodes) = 1$ **then**

43:     $MR \leftarrow CurrentLevelNodes[0]$         $\triangleright$ The single remaining node is the Merkle Root

44: **else**               $\triangleright$ Handles $N = 0$ and $N_{pow2} = 0$, resulting in an empty $CurrentLevelNodes$

45:     $MR \leftarrow H(\texttt{""})$   $\triangleright$ Define Merkle Root for an empty set of tokens, e.g., hash of empty string

46: **end if**

47: **return** $MR$

---

### F.3 WORKFLOW OF CoIn

Algorithm 4 illustrates the complete verification procedure of CoIn.

---

**Algorithm 4** Multi-Round Verification in `CoIn`

---

**Require:** COLA Response (containing reasoning blocks $\mathcal{B}_{\text{total}}$ and final answer $A$)
**Require:** Fraction $\gamma$ of blocks for initial verification (e.g., 0.1)
**Require:** Pre-trained matching heads $\text{MH}_{\text{tb}}(\cdot, \cdot)$, $\text{MH}_{\text{ba}}(\cdot, \cdot)$
**Require:** Embedding function $\text{Embd}(\cdot)$
**Require:** Verification threshold $\tau$
**Ensure:** Audit decision: "Successful" or "COLA Flagged for Inflation"
    *// Initialization*
1:  $\mathcal{B}_{\text{unverified}} \leftarrow \mathcal{B}_{\text{total}}$
2:  $\mathcal{B}_{\text{verified}} \leftarrow \emptyset$
3:  *audit_successful* $\leftarrow$ **false**
4:  *all_blocks_audited* $\leftarrow$ **false**
    *// Initial round of verification*
5:  Select an initial set of blocks $\mathcal{B}_{\text{current\_round}} \subseteq \mathcal{B}_{\text{unverified}}$ of size $\lceil \gamma \cdot |\mathcal{B}_{\text{total}}| \rceil$
6:  **if** $\mathcal{B}_{\text{current\_round}}$ is empty **and** $|\mathcal{B}_{\text{total}}| > 0$ **then**
7:     $\mathcal{B}_{\text{current\_round}} \leftarrow$ one randomly selected block from $\mathcal{B}_{\text{unverified}}$
8:  **end if**
9:  **while not** *audit_successful* **and not** *all_blocks_audited* **do**
10:     **if** $\mathcal{B}_{\text{current\_round}}$ is empty **then**
11:         *all_blocks_audited* $\leftarrow$ **true**           ▷ No more blocks to check
12:         **goto** *FinalDecision*
13:     **end if**
14:     *round_scores* $\leftarrow []$           ▷ Initialize as an empty list/array
15:     **for** each block $B_j \in \mathcal{B}_{\text{current\_round}}$ **do**
16:         Randomly select a subset of reasoning tokens $\{r_i\}_{i=1}^{k}$ from $B_j$ (e.g., 10)
17:         $E_{\text{tokens}} \leftarrow \text{AVG}(\{\text{Embd}(r_i)\}_{i=1}^{k})$     ▷ Average embedding of selected tokens
18:         $E_{\text{block}} \leftarrow \text{Embd}(B_j)$
19:         $E_{\text{answer}} \leftarrow \text{Embd}(A)$
20:         $S_{tb} \leftarrow \text{MH}_{\text{tb}}(E_{\text{tokens}}, E_{\text{block}})$     ▷ Tokens-to-Block score
21:         $S_{ba} \leftarrow \text{MH}_{\text{ba}}(E_{\text{block}}, E_{\text{answer}})$     ▷ Block-to-Answer score
22:         Add pair $(S_{tb}, S_{ba})$ to *round_scores*
23:         `CoIn` performs Merkle Proofs on selected tokens in $B_j$ (verification of token integrity)
24:     **end for**
25:     $\mathcal{B}_{\text{verified}} \leftarrow \mathcal{B}_{\text{verified}} \cup \mathcal{B}_{\text{current\_round}}$
26:     $\mathcal{B}_{\text{unverified}} \leftarrow \mathcal{B}_{\text{unverified}} \setminus \mathcal{B}_{\text{current\_round}}$
27:     *verifier_decision* $\leftarrow \text{VERIFIER}(\textit{round\_scores}, \tau)$    ▷ Verifier can be rule-based or learning-based
28:     **if** *verifier_decision* = Accept **then**
29:         *audit_successful* $\leftarrow$ **true**
30:     **else**
31:         **if** $\mathcal{B}_{\text{unverified}}$ is empty **then**
32:             *all_blocks_audited* $\leftarrow$ **true**
33:         **else**
34:             Select one new random block $B_{\text{next}}$ from $\mathcal{B}_{\text{unverified}}$
35:             $\mathcal{B}_{\text{current\_round}} \leftarrow \{B_{\text{next}}\}$     ▷ Next round verifies this single block
36:         **end if**
37:     **end if**
38: **end while**
    *FinalDecision:*
39: **if** *audit_successful* **then**
40:     **return** "Audit Successful"
41: **else**
42:     **return** "COLA Flagged for Token Inflation"     ▷ User may request COLA to justify charges
43: **end if**

44: **function** VERIFIER(*scores_list*, $\tau$)           ▷ Example: Rule-based verifier
45:     **if** *scores_list* is empty **then return** "Reject"
46:     **end if**
47:     *avg_S_tb* $\leftarrow$ average of all $S_{tb}$ in *scores_list*
48:     *avg_S_ba* $\leftarrow$ average of all $S_{ba}$ in *scores_list*
49:     **if** *avg_S_tb* $> \tau$ **and** *avg_S_ba* $> \tau$ **then**
50:         **return** "Accept"
51:     **else**
52:         **return** "Reject"
53:     **end if**         ▷ Alternatively, a learning-based verifier (e.g., DeepSets) could be used here.
54: **end function**

---

# G   DETECTION PERFORMANCE OF CoIn

We show the comparison of the two verifiers and the impact of $\tau$ under different block sizes, as shown in Figure 7,8,9,10

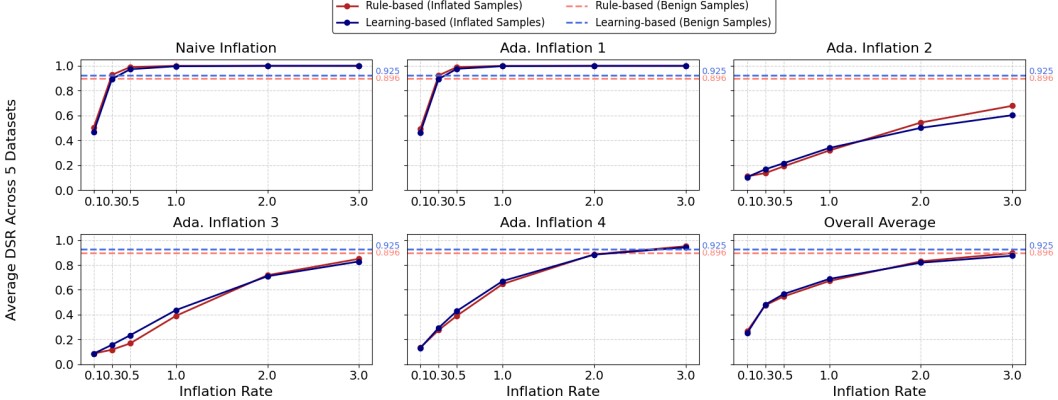

Figure 7: Performance of CoIn across different inflation methods and verifiers (Block Size = 512). The red lines and the blue lines represent the DSR of rule-based verifier and learning-based verifier, respectively. For a fair comparison, we set the threshold $\tau = 0.5$ for learning-based verifier, and $\tau = 0.6$ for rule-based verifier.

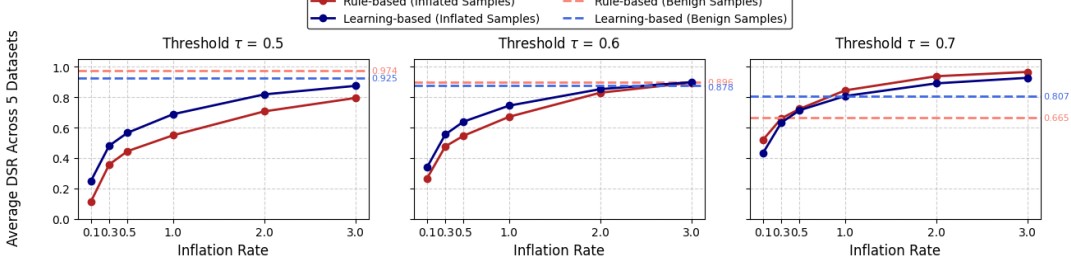

Figure 8: Impact of threshold $\tau$ on DSR (Block Size = 512).

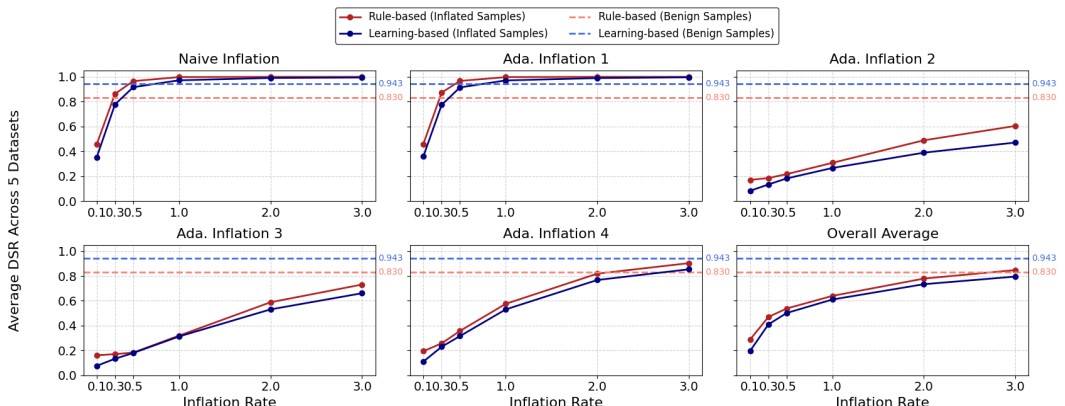

Figure 9: Performance of `CoIn` across different inflation methods and verifiers (Block Size = 1024). The red lines and the blue lines represent the DSR of rule-based verifier and learning-based verifier, respectively. For a fair comparison, we set the threshold $\tau = 0.5$ for learning-based verifier, and $\tau = 0.6$ for rule-based verifier.

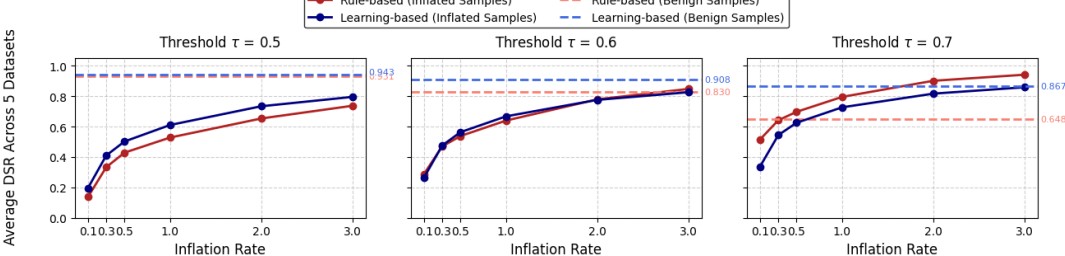

Figure 10: Impact of threshold $\tau$ on DSR (Block Size = 1024).

## H   Prompts Used in Discussion Section

Prompt 11 is used to explore the question "Can the original text be recovered from the tokens and embeddings exposed by COLA?", while Prompt 12 is used to explore "How difficult is the dataset we constructed?".

---

**Prompt for Reconstructing Hidden Reasoning Passage**

**Reconstructing Hidden Reasoning Prompt**
You are an expert in natural language reasoning and semantic retrieval.
Your task is to help recover a semantically meaningful and logically connected hidden passage that bridges a <Question> and an <Answer>.
This passage has been lost, but we know it is semantically related to both the <Question> and the <Answer>, and lies between them.

Given a – **<Question>**:
{question}

And the – **<Answer>**:
{answer}

We also know that some tokens from the original passage are still visible:
{sampled_token_text}

And we retrieved related documents from Wikipedia using the embedding of the original passage:
{retrieved_rag_docs}

Now, please help recover the most likely content of the hidden passage.
Return your answer strictly in the following JSON format:

```
 \\recovered_json {
   "recovered_text": "<your reconstructed passage here>"
 }
```

---

Figure 11: Prompt for Recovering a Hidden Reasoning Passage Using Question, Answer, Token Clues and Retrieved Wikipedia Documents.

**Prompt for Evaluating Reasoning Passage Relevance**

**Evaluating Reasoning Process Prompt**
You are a logical reasoning analyst.

Given a final answer and a randomly selected text passage, your task is to assess whether the text passage represents a reasoning process that leads to or supports the final answer.

The passage may or may not be relevant to the answer.
Your task is **not** to verify factual correctness, but to determine whether the passage semantically or logically connects to the answer and explains or justifies it in any meaningful way.

**Random Text Passage**:
{reason}

**Final Answer**:
{answer}

Please answer the following questions:
1. Is the text passage a plausible reasoning process that leads to the final answer?
2. Does it provide logical or semantic justification for the answer?

Respond in the following JSON format:

```
  \\reasoning_assessment
{
   "is_reasoning_process": true/false,
   "justification": "<your brief explanation of why the passage is or isn't
a reasoning process for the answer>"
}
```

Figure 12: Prompt for Judging Whether a Block Supports or Explains a Final Answer.

# I CAN THE ORIGINAL TEXT BE RECOVERED FROM THE TOKENS AND EMBEDDINGS EXPOSED BY COLA?

## I.1 DIRECT RECONSTRUCTION FROM EMBEDDINGS

**Objective.** Our framework aims to protect the provider's intellectual property by preventing large-scale harvesting of reasoning traces for distillation or reverse-engineering. We do *not* target the protection of isolated reasoning fragments, which are often trivially inferable from final answers or API summaries and are thus not considered sensitive. The real risk arises only when many reasoning steps are aggregated, enabling low-cost imitation of proprietary reasoning strategies.

**Method.** To assess direct reconstruction risk, we adopt the hypothesis–correction embedding inversion approach Vec2Text Morris et al. (2023) with the publicly released `inversion_model` and `corrector_model` trained for OpenAI's `text-embedding-ada-002`. We follow the default configuration (50 update steps; sequence beam width = 4).

**Findings.** As shown in Table 5, partial recovery is possible only when the block size is small ($\leq 64$). Consistent with prior observations Morris et al. (2023), reconstruction accuracy degrades sharply as length increases. In high-reasoning-load domains such as *math* and *code*, the reconstruction task becomes especially difficult. At our framework's minimum block size of 256, the attack completely fails. Even when BLEU/F1 appear non-trivial, the recovered text typically exhibits severe semantic drift, rendering it unsuitable for malicious distillation or dataset construction.

| Block Size | Code | | General | | Math | | Medical | |
|---|---|---|---|---|---|---|---|---|
| | BLEU | Token F1 | BLEU | Token F1 | BLEU | Token F1 | BLEU | Token F1 |
| 16 | 31.08 | 0.6913 | 87.70 | 0.9292 | 43.83 | 0.7467 | 73.88 | 0.8845 |
| 32 | 22.98 | 0.5433 | 57.80 | 0.7828 | 15.06 | 0.5123 | 48.12 | 0.7195 |
| 64 | 12.31 | 0.4437 | 32.44 | 0.6412 | 12.62 | 0.4256 | 31.88 | 0.6291 |
| 128 | 7.98 | 0.3206 | 14.34 | 0.4816 | 9.84 | 0.3441 | 16.82 | 0.4928 |
| 256 | 3.24 | 0.2603 | 4.37 | 0.3740 | 4.07 | 0.2617 | 5.73 | 0.3801 |
| 512 | 0.46 | 0.2048 | 0.68 | 0.2989 | 0.67 | 0.2016 | 0.78 | 0.3101 |

Table 6: Direct reconstruction from embeddings using Vec2Text Morris et al. (2023) (50 update steps; beam width 4). Recovery drops rapidly with length; at block size 256 (our minimum), attacks effectively fail.

**Example (Max Token Length = 64).** Even with token-level overlap, the semantics are heavily distorted:

```
Original:
formula OH2 = 9R2 - (a2 + b2 + c2). If R=4, then 9R2=9*16=144.
The sum of squares of the sides: a2 + b2 + c2. From earlier, a=BC=55, so

Predicted:
formula: OH(a2+b2+c2) = R(a2+b2+c2) = R(a2+b2+c2) = R(a2+b2+c2) ...
= R(a2+b2+c2) = R(a2+b2+c2) = R(a2+b2+c2) = 55.
If a square has sides, then R(a2+b2+c2) = 9. If a square has sides, then
```

This illustrates that despite moderate BLEU/F1, the reconstructed content diverges in meaning and cannot serve as a basis for malicious distillation or dataset construction.

## I.2 LM-BASED RECONSTRUCTION

During the verification process in `CoIn`, COLA leaks a certain number of block embeddings and tokens within the blocks to `CoIn`. To quantify the impact of such leakage, we assume a malicious `CoIn` leverages an RAG system to retrieve documents highly similar to the exposed embeddings and tokens, then feeds all retrieved information into an LLM to reconstruct the original content. The design and further details are provided in Appendix H. We randomly selected 100 samples from a mathematical dataset. We evaluated the similarity between the reconstructed blocks and the original

Table 7: Similarity Between Blocks Reconstructed by `CoIn` and Real Blocks.

| Metric | Block Size $\beta$ | | |
|---|---|---|---|
| | 256 | 512 | 1024 |
| EmbedSim | 0.65 | 0.66 | 0.75 |
| BLEU | 0.04 | 0.05 | 0.03 |
| ROUGE-L | 0.23 | 0.25 | 0.24 |
| BERTScore | 0.83 | 0.83 | 0.84 |

ones using embedding similarity, BLEU score Papineni et al. (2002), ROUGE-L Lin (2004) , and BERTScore Zhang et al. (2019). As shown in Table 7, we observe that a high BERTScore/EmbedSim combined with low BLEU/ROUGE indicates that the LLM successfully preserves the core semantics, while significantly differing from the real block in terms of surface expression and syntactic structure.

## J  DEFENDING AGAINST REPETITION-BASED TOKEN INFLATION ATTACKS

### J.1  THREAT CHARACTERIZATION

Beyond inflation attacks that `CoIn` primarily targets, an important class of adversarial strategies involves **repetition-based manipulation**. In such cases, a dishonest provider may artificially inflate the number of tokens by (1) duplicating subsequences of the reasoning text, or (2) appending variants of the original content generated through permutation or rewriting with small LLMs. These strategies differ from simple fabrication in that they preserve surface plausibility, but nonetheless undermine fair billing.

It is important to note that `CoIn` does not claim to defend against all advanced manipulations. Achieving complete security is difficult, and—as is the case across many trustworthy AI domains such as jailbreak defense, backdoor detection, and federated learning—robustness is typically achieved through iterative adversarial refinement. The value of `CoIn` lies in providing a balanced foundation that can be extended with additional defensive layers.

### J.2  DETECTING DIRECT REPETITION

For the case of direct duplication, such as repeating hidden sequences or copying token fragments, `CoIn` can be extended with subtree equivalence checks on Merkle tree substructures. During an audit, the user may request all hash values at a specific level $n$ of the Merkle tree. If repeated content exists, identical hash values will appear at sibling positions. This is because identical hashes at sibling nodes necessarily imply that the corresponding subtrees are structurally the same, which in turn means that the underlying token sequences are repeated. Since the provider must still produce valid Merkle proofs, these hash values necessarily correspond to real content; any attempt to falsify them would invalidate the proof.

In addition, as described in the `CoIn` design, the user retains the right to request the original reasoning text whenever suspicious billing arises. Once disclosed, repeated segments are trivially observable, making deception practically infeasible.

### J.3  ADDRESSING PERMUTED OR REWRITTEN VARIANTS

A more sophisticated attack involves appending **permuted or rewritten versions** of the original output, possibly produced by a small auxiliary LLM. While such strategies fall outside `CoIn`'s initial threat model, since they require non-trivial computation rather than near-zero-cost inflation, they remain detectable through **redundancy analysis**.

To demonstrate this, we construct a controlled experiment:

- Let $R_{\text{ori}}$ denote the original reasoning text.
- Generate an alternative sequence $R_{\text{llm}}$ using a small LLM (Qwen/Qwen3-4B-Instruct-2507 Yang et al. (2025a)).
- Form the combined text $R_{\text{combined}} = R_{\text{ori}} \oplus R_{\text{llm}}$, which yields 100% inflation.

We then compute **normalized n-gram redundancy rates**. For a token sequence of length $L$ and an n-gram window size $n$, let $D_n$ denote the number of distinct n-grams appearing more than once. The redundancy rate is defined as:

$$\text{Redundancy}(n) = \frac{D_n}{L - n + 1}.$$

Intuitively, this metric captures how much repetition exists in a sequence: the higher the value, the more internally redundant the text is. Normalization by $(L - n + 1)$ eliminates the influence of text length, allowing fair comparisons across different samples.

## J.4 EMPIRICAL RESULTS

The redundancy rates for different domains and n-gram sizes are summarized in Table 8. The results show that $R_{\text{combined}}$ consistently exhibits much higher redundancy than either $R_{\text{ori}}$ or $R_{\text{llm}}$. Even when using LLM rewriting, substantial overlaps remain between the original and fabricated text. These redundancy patterns can be effectively detected by the Merkle subtree integrity check, thereby enabling `CoIn` to flag such manipulations.

| Source | n-gram size | $R_{\text{ori}}$ | $R_{\text{llm}}$ | $R_{\text{combined}}$ |
|---|---|---|---|---|
| Code | 8 | 4.23 | 6.84 | 18.10 |
| | 16 | 1.00 | 4.40 | 17.23 |
| | 32 | 0.29 | 3.48 | 16.60 |
| General | 8 | 0.24 | 0.47 | 9.32 |
| | 16 | 0.04 | 0.14 | 8.95 |
| | 32 | 0.01 | 0.07 | 8.72 |
| Math | 8 | 5.21 | 7.28 | 12.18 |
| | 16 | 1.12 | 4.43 | 9.78 |
| | 32 | 0.13 | 3.45 | 8.90 |
| Medical | 8 | 1.40 | 3.45 | 9.27 |
| | 16 | 0.20 | 2.15 | 7.91 |
| | 32 | 0.03 | 1.65 | 7.14 |

Table 8: Normalized redundancy rates of original text ($R_{\text{ori}}$), LLM-generated variant ($R_{\text{llm}}$), and combined text ($R_{\text{combined}}$) across domains. Higher values indicate stronger repetition.

In summary, `CoIn`'s design can be naturally extended to resist repetition-based token inflation attacks:

- **Direct duplication** is detectable via subtree hash checks.
- **Permuted or rewritten variants** can be identified through redundancy analysis combined with Merkle tree verification.

Although such strategies are outside the strict scope of `CoIn`'s original threat model, these results illustrate the framework's extensibility. As with other trustworthy AI systems, we expect robustness to improve through iterative adversarial refinement and the integration of complementary defenses.

