# OpenReview forum: "CoIn: Counting the Invisible Reasoning Tokens in Commercial Opaque LLM APIs"
_ICLR.cc/2026/Conference — ICLR 2026 Conference Withdrawn Submission_

### Official Review · Reviewer_Nmj6 · 2025-10-16

**Soundness:** 2
**Presentation:** 3
**Contribution:** 2
**Rating:** 4
**Confidence:** 4

**Summary:**

This paper addresses a critical transparency issue in commercial reasoning-enhanced LLM APIs (termed COLA - Commercial Opaque LLM APIs) where providers conceal intermediate reasoning traces while billing users for invisible reasoning tokens. The authors formalize the problem of "token count manipulation" and propose CoIn, a verification framework that combines token quantity verification using Merkle hash trees and semantic validity verification using embedding-based matching heads. The framework enables third-party auditing of hidden tokens without exposing proprietary content. Experiments demonstrate that CoIn can detect various token inflation strategies with detection success rates reaching 94.7% under certain conditions, while maintaining low exposure rates of reasoning content.

**Strengths:**

- The paper identifies and formalizes an important emerging problem in commercial LLM services, namely the lack of transparency in billing for invisible reasoning tokens, which becomes increasingly relevant as reasoning models proliferate.
- The technical approach cleverly combines cryptographic verification (Merkle trees) with semantic validation (embedding-based matching), providing both quantity and quality checks on hidden tokens.
- The experimental evaluation spans multiple domains (medical, code, math, general) and systematically tests various inflation strategies, demonstrating thoroughness in the experimental design.
- The paper provides extensive supplementary materials including detailed algorithms, dataset construction procedures, and code availability, supporting reproducibility.
- The framework maintains relatively low exposure rates (less than 40% embedding exposure and less than 4% token visibility as stated in the abstract) while achieving reasonable detection performance on certain attack types.

**Weaknesses:**

- The threat model contains a fundamental contradiction by requiring malicious providers to actively cooperate with the auditing process, including generating embeddings using a fixed model and providing Merkle proofs, which a truly adversarial provider would simply refuse to do.
- The "near-zero cost" assumption for token inflation is unrealistic because providers could easily use small language models (e.g., 1-4B parameter models) to generate semantically plausible fake reasoning at minimal computational cost, bypassing the semantic validity checks.
- Detection performance degrades substantially for adaptive attacks, with some scenarios achieving only 74.7% DSR (Table 4) and the framework maintaining only 40.1% success probability when 10% of tokens are forged, indicating limited robustness against sophisticated adversaries.
- The semantic validity verification relies on embeddings that could be manipulated by a malicious provider using a different embedding model than specified, and the paper does not address how to verify that COLA actually uses the designated embedding model.
- The paper acknowledges in Appendix B that important attack vectors remain unaddressed, including LLM-based rewriting, cascade inference, and speculative sampling, which significantly limits the practical applicability of the framework.
- The evaluation datasets consist primarily of artificially constructed inflation attacks rather than realistic adversarial strategies informed by actual provider incentives and capabilities, potentially overstating the framework's effectiveness.
- The trade-off between detection capability and block size (Table 2) creates a tension where smaller blocks improve detection but increase COLA's exposure risk, and larger blocks reduce exposure but substantially decrease detection success rates for adaptive inflation (as shown in Appendix G).

**Questions:**

1. How do you envision enforcing provider cooperation in practice? What mechanism would prevent a malicious COLA from simply refusing to participate in the CoIn auditing process or providing manipulated data?
2. Could you provide a more concrete cost analysis comparing the computational expense of generating plausible fake reasoning using small LMs (e.g., 1-4B parameter models) versus the potential revenue from token inflation? This would help assess whether the "near-zero cost" assumption is realistic.
3. In Table 4, the Tokens-to-Block verification struggles with Adaptive Inflation 2 (74.7% DSR). Could you explain why this particular attack is so challenging and what modifications to CoIn might improve detection?

---

### Official Review · Reviewer_yfsh · 2025-10-31

**Soundness:** 1
**Presentation:** 3
**Contribution:** 1
**Rating:** 2
**Confidence:** 4

**Summary:**

This work address a transparency gap between LLM API users and service providers: API service providers may manipulate token count and hence manipulate bills which are typically calculated by token count. The authors proposes CoIn, a framework to audit quantity and semantic validity of hidden tokens. The authors have conducted extensive experiments to evaluate the proposed pipeline.

**Strengths:**

The  paper is well-written and easy to follow.

**Weaknesses:**

1. Validity of the threat model
    - In practice, other challenges related to honest billing may arise; for instance, COLA may charge for tokens differently depending on a prompt cache-hit status. Consequently, the scope of this paper is limited.
    - This paper ignores the fact that COLA may reveal a summary (instead of full) of CoT traces to users. The reviewer believes that the work should include this information for semantic validity verification.
    - Another concern is that, is there any user report that raises the mentioned issue in the paper? Any online discussion, report, preprints are welcome.
2. Method soundness.
    - The reviewer is concerned that a simple baseline method to bypass quantity and semantic filter: COLA uses a smaller model response instead of the original model. It is believed that such baseline method would not fail, as long as the API serving process keeps opaque.
3. Implications.
- The paper must include a section dedicated to discussing the implications of the proposed CoIn framework. This framework requires the COLA  to report the generated embeddings for all reasoning tokens. Crucially, the integrity of this implementation cannot rely solely on COLA itself. A trusted third party—similar to a Certificate Authority (CA) in web security—must be established to ensure that COLA correctly and honestly executes the required data generation pipeline. The authors are strongly encouraged to discuss how governments or industry leaderscould establish and regulate such a necessary oversight mechanism.

**Questions:**

Please refer to the “Weakness” part above.

---

### Official Review · Reviewer_dXDo · 2025-11-01

**Soundness:** 1
**Presentation:** 2
**Contribution:** 1
**Rating:** 2
**Confidence:** 4

**Summary:**

This paper introduces the CoIn (Counting the Invisible) framework to audit Commercial Opaque LLM APIs that hide reasoning tokens but bill for them. CoIn acts as a third-party service, employing a Verifiable Hash Tree to check the integrity of the total token quantity. It also utilizes a lightweight Semantic Matching Head to detect the injection of irrelevant or fabricated tokens.

**Strengths:**

- The methodology is straightforward and easy to understand.

- The paper is well-structured and clearly organized.

**Weaknesses:**

- The research question presented in this work lacks sufficient practical grounding. If a commercial provider of LLM API intends to increase user payment costs, directly raising the price of each token would be a more straightforward and effective approach. In contrast, increasing charges by inflating the consumption of reasoning tokens appears neither practical nor necessary from a business perspective.

- The Semantic Validity Verification relies on consistency checks between reasoning tokens and their block/answer tokens. However, the mechanism appears vulnerable if the Commercial Opaque LLM APIs choose to simply duplicate or repeat a subset of existing reasoning tokens. While the experiments include sampling from various sources (reasoning, answer, and prompt tokens), the paper lacks a dedicated analysis or a clear result specifically addressing the detection success rate against this particular self-duplication/repetition attack originating only from the reasoning tokens.

- While Quantity Verification effectively ensures count integrity, it has inherent limitations, as any inflation introduced during Merkle Tree construction would remain undetected. Moreover, inflating the token count by duplicating existing valid reasoning tokens could also bypass the Semantic Validity Verification system.

- The font sizes of Figure 2 appear too small; please increase them to improve overall readability.

**Questions:**

- How the proposed token inflation types were derived? Are they supported by real-world investigations or empirical evidence?


- Have the authors considered potential methods to circumvent the proposed detection mechanism? What is the estimated success rate of such attacks? How could the defense be strengthened in response?

- Why does the experiment lack proper baseline methods, and why does the rule-based approach resemble an ablation study?

- What is the theoretical basis underlying the proposed defense mechanism? Specifically, the paper only considers a fully invisible reasoning chain, whereas real-world scenarios often involve partially or fully visible ones. How should attack and defense strategies be adapted in such cases?

- Is there any evidence that commercial LLM API providers actually engage in reasoning token inflation in practice? Without such real-world instances, the practical relevance of the work remains unclear.

- The paper uses low semantic consistency (i.e., low relevance) as a proxy for low reasoning token quality (i.e., fabricated or injected tokens). Could the authors justify the underlying assumption that low semantic relevance necessarily implies low quality or malicious intent?

---

### Official Review · Reviewer_gqW9 · 2025-11-03

**Soundness:** 3
**Presentation:** 3
**Contribution:** 2
**Rating:** 4
**Confidence:** 2

**Summary:**

disclaimer: this work is a bit outside my technical expertise so i set the confidence level as 2.

This paper proposes CoIn, a two-fold verification framework to detect token inflation in opaque reasoning LLM APIs. On one hand, by constructing hash-tree and applying merkle proof, users can identify if a queried token is contaminated. On the other hand, block semantic verification serves another solution to detect if the reasoning contents are consistent from each other.

**Strengths:**

1. this work raises awareness of a critical problem: token inflation, which is new after the release of reasoning LLMs such as GPT o1.

2. this work proposes two novel approaches for token inflation detection, from hash-tree construction to semantic match.

**Weaknesses:**

1. the effectiveness of the proposed solution is a bit limited: according to figure 3, under most of the scenarios IRs have to be at least 1.0 for the detection accuracy to be decent. particularly, in  Ada. Inflation 2, no meaningful performance is achieved until IR=3. questioning the solution's effectiveness when only a fraction of tokens are added.

2. overall the experimental setting is a bit hypothetical: (a) token inflation scenarios as presented in table 1 are a bit too synthetic, Ada. Inflation 2 seems the most realistic case to me. (b) Furthermore, the embedding generation and hash-tree construction may add further costs to the providers, which challenges its practicality. this cost (storage, compute) analysis should be performed beyond figure 5.

3. the authors should add more experiment results using more reasoning LLMs, e.g. Qwen3-235B, GPT-OSS (if possible, since it's released in early August).

**Questions:**

1. if i understand correctly, figure 3 presents results for TOKEN QUANTITY VERIFICATION, which doesn't involve rule-based or learning-based matching heads etc (used in semantic matching), why does it include two verifier types?

2. figure 6, why is the misclassification rate for no-inflation is that high, while  Ada. Inflation 2 and 3 are actually lower? this appears counter-intuitive to me. also, a minor point, maybe i missed this, is figure 6 ever mentioned in the main text?

3. does CoIn performance depend on tokenizer types?

---

### Note · Authors · 2026-01-12

**Comment:**

I have read and agree with the venue's withdrawal policy on behalf of myself and my co-authors.

**Withdrawal Confirmation:**

I have read and agree with the venue's withdrawal policy on behalf of myself and my co-authors.